# Efficacy of an Internet-Based Program to Promote Physical Activity and Exercise after Inpatient Rehabilitation in Persons with Multiple Sclerosis: A Randomized, Single-Blind, Controlled Study

**DOI:** 10.3390/ijerph17124544

**Published:** 2020-06-24

**Authors:** Peter Flachenecker, Anna Karoline Bures, Angeli Gawlik, Ann-Christin Weiland, Sarah Kuld, Klaus Gusowski, René Streber, Klaus Pfeifer, Alexander Tallner

**Affiliations:** 1Neurological Rehabilitation Center Quellenhof, 75323 Bad Wildbad, Germany; BuresA@dbknb.de (A.K.B.); Ann-Christin.Weiland@Sana.de (A.-C.W.); Klaus.Gusowski@sana.de (K.G.); 2Department of Neurology, University of Würzburg, 97080 Würzburg, Germany; 3Department of Health & Social Psychology, German Sport University Cologne, 50933 Cologne, Germany; A.Gawlik@dshs-koeln.de; 4Department of Sport Science and Sport, University of Erlangen-Nuremberg, 91058 Erlangen, Germany; Rene.Streber@fau.de (R.S.); Klaus.Pfeifer@fau.de (K.P.); 5Faculty of Economics, University of Bayreuth, 95447 Bayreuth, Germany; Sarah.Kuld@gmx.de

**Keywords:** multiple sclerosis, rehabilitation, fatigue, quality of life, walking, physical activity, exercise, online systems, internet-based intervention, health behavior

## Abstract

Background: Multimodal rehabilitation improves fatigue and mobility in persons with multiple sclerosis (PwMS). Effects are transient and may be conserved by internet-based physical activity promotion programs. Objective: Evaluate the effects of internet-based physical activity and exercise promotion on fatigue, quality of life, and gait in PwMS after inpatient rehabilitation. Methods: PwMS (Expanded Disability Status Scale (EDSS) ≤ 6.0, fatigue: Würzburg Fatigue Inventory for Multiple Sclerosis (WEIMuS) ≥ 32) were randomized into an intervention group (IG) or a control group (CG). After rehabilitation, IG received 3 months of internet-based physical activity promotion, while CG received no intervention. Primary outcome: self-reported fatigue (WEIMuS). Secondary outcomes: quality of life (Multiple Sclerosis Impact Scale 29, MSIS-29), gait (2min/10m walking test, Tinetti score). Measurements: beginning (T0) and end (T1) of inpatient rehabilitation, 3 (T2) and 6 (T3) months afterwards. Results: 64 of 84 PwMS were analyzed (IG: 34, CG: 30). After rehabilitation, fatigue decreased in both groups. At T2 and T3, fatigue increased again in CG but was improved in IG (*p* < 0.001). MSIS-29 improved in both groups at T1 but remained improved at T2 and T3 only in IG. Gait improvements were more pronounced in IG at T2. Conclusions: The study provides Class II evidence that the effects of rehabilitation on fatigue, quality of life, and gait can be maintained for 3–6 months with an internet-based physical activity and exercise promotion program.

## 1. Introduction

Fatigue is a common and disabling symptom in patients with multiple sclerosis (PwMS) and contributes to reduced activities of daily living as well as to the loss of work ability, thus negatively affecting quality of life [1]. Despite its enormous impact on PwMS, fatigue treatment options are scarce [2], and a large proportion of PwMS suffering from fatigue remain untreated [3]. Several studies have consistently shown that physical exercise with endurance and resistance training is effective at ameliorating the devastating consequences of fatigue [4,5], and a systematic review of systematic reviews found strong evidence for the reduction of fatigue with exercise-based educational programs [6]. In this regard, multimodal rehabilitation offers the opportunity not only to administer different types of endurance and resistance training and to motivate PwMS to maintain or increase physical activity levels, but also to inform and educate them that continuously done exercise has a number of positive effects on various symptoms of MS [7] and that it can be performed without concerns with regard to side effects or deterioration of the disease [8]. 

After discharge from rehabilitation, though, patients are faced with the challenge of having to implement adequate physical activity and exercise levels into their daily routines. Cessation of therapeutic support and guidance frequently leads to a loss of motivation and subsequent physical inactivity after rehabilitation, causing a gradual decline in health [9]. Thus, home-based, behavior-oriented programs that deliver therapeutic support after rehabilitation are needed to maintain positive results. To our knowledge, there are no such home-based aftercare programs in Germany that are specifically designed for PwMS. In this regard, the internet is a useful tool to deliver disease-specific and symptom-oriented exercise, as well as physical activity promotion that is location independent, widely available, and easily accessible for PwMS [10]. 

In a previous study, we showed that internet-based endurance and resistance training that was specifically developed for PwMS, individually tailored towards their needs, and administered online by exercise therapists improved physical activity, muscle power, and respiratory function [11]. Whether this approach may also be useful to maintain the effects of inpatient rehabilitation and to ameliorate the burden of fatigue in the long term is not known. Therefore, we aimed at investigating the effects of a 3 month, internet-based program to promote physical activity and exercise on the sustainability of the effects of inpatient rehabilitation over a period of 3 and 6 months.

## 2. Patients and Methods

All patients admitted to inpatient rehabilitation at the Neurological Rehabilitation Center Quellenhof between August 2015 and May 2016 were considered eligible for the study. Inclusion criteria were (1) diagnosis of MS according to 2005 McDonald criteria [12]; (2) age ≥ 18 years; (3) EDSS (Expanded Disability Status Scale) [13] ≤ 6.0; (4) presence of fatigue, as indicated by a Würzburg Fatigue Inventory for Multiple Sclerosis (WEIMuS) score ≥ 32 [14,15]; (5) willingness to undergo an outpatient visit after 3 months and to participate in a postal survey after 6 months; and (6) internet access and basic computer knowledge. Patients were excluded if they (1) had a relapse and/or had received corticosteroids within 30 days before inclusion; (2) suffered from cognitive deficits, severe hand dysfunction, and/or serious cardiovascular disease (heart failure, cardiac arrhythmia, aortic stenosis, instable hypertension); and (3) had already performed regular endurance (≥2/week) and/or resistance training (≥1/week). Written informed consent was obtained from all patients. The study was approved by the local ethics committee (Landesärztekammer Baden-Württemberg, ref 2009-099-f) and registered at “Deutsches Register für Klinische Studien (DRKS)”, No. DRKS00020291.

Patient groups and internet-based intervention: After written informed consent, patients were randomly allocated to an intervention group (IG) or a control group (CG) using block randomization with a block size of n = 10. Both groups underwent the usual, goal-oriented, specifically tailored multimodal inpatient rehabilitation program. After discharge, IG received an internet-delivered, behavior-oriented exercise and physical activity promotion program for 3 months. The program was organized and delivered by the Department of Sport Science and Sport (DSS) of the University of Erlangen-Nuremberg, Germany.

The program aimed at increasing motivational and volitional determinants as well as necessary competences for a self-determined, physically active lifestyle. It was theoretically based on the model of physical-activity-related health competence [16,17] and on the self-determination theory [18], and it integrated several techniques of behavioral change and the concept of motivational interviewing [19]. A detailed description of the methodology is given elsewhere [20]. A half-day educational seminar conducted at the end of inpatient rehabilitation by staff from DSS was the starting point for two integrative program components: (1) web- and telephone-based, behavior-oriented physical activity coaching with one individual and four group sessions, and (2) an individual exercise prescription in a one-to-one approach using a specialized, browser-based software solution (motionNET E-training, Nuremberg, Germany). Participants used the software to document their exercises and to plan their activities and sessions in a physical activity diary. Exercise therapists used patient feedback and exercise parameters (ratio of perceived exertion, heart rate) to supervise and manage exercises and activities. Perceived exertion was rated on the Borg Scale ranging from 6 (no exertion) to 20 (maximum exertion) for each exercise. The communication with patients took place via a built-in messenger or by e-mail, telephone, or video conference. Participants determined their exercise regime in consultation with their therapists, according to their individual goals and health situation. Individual exercise prescription was based on general recommendations for strength training (6–8 exercises for the major muscle groups, 1–2 times per week) and endurance training (free choice of activity, 10–60 min, 1–2 times a week). The recommendation for exercise intensity was light to moderate, corresponding to a Borg Scale score between 11 and 15. There was no standardized warmup for training sessions. All exercises could easily be performed at home without expensive equipment. Therapists could choose from a catalog with 220 exercises (strength, endurance, core stability, balance, and flexibility) that accounted for varying fitness levels and functional limitations. This exercise catalog was specifically composed for PwMS and established in previous studies [11,21]. For severely affected patients, the catalog included specifically adapted exercises in sitting, lying, or kneeling positions and exercise instructions included precautions to avoid falling or stepping. Prescription was performed on the basis of an extensive anamnesis by two exercise therapists with extensive rehabilitation experience. The training was performed over a period of 3 months and started directly after discharge from inpatient rehabilitation. Adherence to the intervention was derived from the electronically documented training sessions and analyzed as average training sessions per week. 

PwMS in the control group (CG) were cared for as usual after discharge. They did not receive any study intervention and were told not to change any of their habits, including physical activity. Outcome parameters were obtained at the same time points as in the IG. From month 3 to month 6 after discharge, neither group received any intervention. Health care utilization between the end of rehabilitation (T1) and 6 months thereafter (T3), including physiotherapy, was neither facilitated nor restricted for IG and CG alike. 

Assessments: Measurements were performed within the first week of inpatient rehabilitation (T0), at the end of rehabilitation (T1), and 3 months (T2) and 6 months thereafter (T3), as illustrated in Table 1. The evaluating assessors were unaware of group allocation, reflecting the single-blind design of the study. Patients were instructed before the assessments not to report whether they had exercised via the internet or not. 

The primary outcome was the subjective dimension of fatigue after 3 months compared to baseline (T0) for IG vs. CG, as measured with the WEIMuS questionnaire (see below). Secondary outcomes were fatigue after 6 months, health-related quality of life (HRQoL) at discharge and after 3 and 6 months, as well as mobility parameters at T1 and T2, compared to baseline, for each group.

Fatigue was assessed with the WEIMuS questionnaire (Würzburg Fatigue Inventory for Multiple Sclerosis) [15]. This self-reported instrument consists of 17 items ranging from “0” to “4”, resulting in a total sum score from 0 to 68, with higher scores indicating higher degrees of fatigue. In a normative study with 161 healthy volunteers, sum scores below 32 fell within the range of 1.5 times the standard deviation and thus, a WEIMuS sum score ≥ 32 was used as the cutoff to indicate clinically relevant fatigue [14]. By using this criterion, 84% of MS patients with fatigue were correctly classified, and only 4% of MS patients without fatigue scored above this value [14]. 

HRQoL was determined with the Multiple Sclerosis Impact Scale 29 (MSIS-29), version 1 [22,23]. This is an MS-specific patient-reported outcome that consists of 29 items in 2 domains (psychological and physiological), which were rated from “1” to “5”, with higher scores indicating lower levels of HRQoL. The raw scores range from 29 to 145 for the total score, from 20 to 100 for the physical subscale, and from 9 to 45 for the psychological subscale. 

Gait and balance measures included the 10 m walking test (10mWT) and the 2 min walking test (2minWT) to assess short- and long-distance walking capacity, as well as the Tinetti score (TS, also known as “Performance-Oriented Mobility Assessment”, POMA) to assess balance. These assessments were carried out by two assessors (A.B., A.W.). The number of steps and the time (in seconds) needed for 10 m with walking speed as fast as possible (10mWT) and the distance (in meters) that could be walked in 2 min (2minWT) were recorded [24]. For TS, different tasks of balance in two domains were rated by the evaluating physiotherapist on a scale from “0” to “2”, resulting in a total sum score from “0” to “28”, with higher scores reflecting higher degrees of balance [25].

Statistics: Demographic characteristics were given as mean (+/− standard deviation), or median and 25%-75% interquartile ranges, respectively. Due to the specified inclusion criteria regarding the primary outcome (WEIMuS fatigue score ≥32 on the 68-point WEIMuS scale), only PwMS at one end of the scale regarding fatigue levels were included. Normal distribution was checked with the Shapiro–Wilk test, and equal variances with the Brown–Forsythe test. Since normal distribution or equal variance was not present in all of the analyzed variables, nonparametric tests were used for all comparisons. Data for primary and secondary outcomes were expressed as median and 25–75% interquartile ranges. Differences between groups (primary outcome) were assessed with the Mann–Whitney rank sum test, differences within both groups (secondary outcomes) were estimated with the Friedman Repeated Measures Analysis of Variance on Ranks with pairwise comparison procedures (Tukey’s test). Results were considered statistically significant for *p* < 0.05. Since this was a pilot study, a sample size calculation could not be performed. Post hoc analysis of the primary outcome revealed a power of 0.925. All analyses were performed per protocol, using the standard software package SIGMAPlot for Windows Version 13.0, Systat Software 2014.

## 3. Results

Out of 584 PwMS admitted to inpatient rehabilitation during the study period, 364 did not fulfill the inclusion and exclusion criteria, and 136 declined participation. Thus, 84 patients were randomly allocated to IG (n = 42) and CG (n = 42). Data from 20 patients were excluded due to different reasons (Figure 1). Thus, the study population consisted of 64 PwMS (34 in IG, 30 in CG). The demographic and baseline characteristics of both groups are given in Table 2. There were no statistically significant differences between IG and CG except for disease duration, which was higher for the IG. 

### 3.1. Primary Outcome: Fatigue (WEIMuS Questionnaire)

Fatigue: Median WEIMuS scores at baseline were higher in IG (indicating a higher degree of fatigue) than in CG but dropped similarly in both groups, indicating the positive effects of inpatient rehabilitation (*p* < 0.001, discharge vs. baseline, Figure 2). At month 3, WEIMuS scores were again increased in CG, without a significant difference at month 6 compared to baseline (Figure 2). In contrast, the WEIMUS scores remained essentially stable in IG at month 3, without a significant difference between month 3 and discharge, and decreased even further at 6 months (*p* < 0.01, month 6 vs. discharge). Compared with baseline, all measurement points in IG showed statistically significant differences (*p* < 0.001 for all comparisons, Figure 2).

With regard to differences, the median improvement in the WEIMuS scores at discharge was essentially similar in IG compared to CG (15.5 vs. 18.0; *p* = 0.28), whereas at both follow-up measurements, the improvements in IG were superior to those obtained in CG (month 3: 16.5 vs. 7.0; month 6: 22.5 vs. 5.5, *p* < 0.001 for each comparison, Table 3). 

### 3.2. Health-Related Quality of Life (MSIS-29)

The time courses for HRQoL as measured with the total MSIS-29 scale were comparable to those of the WEIMuS scores (Figure 3). Again, inpatient rehabilitation led to a comparable, significant improvement in both groups. After 3 months, the HRQoL in CG returned to values similar to those at baseline. The improvements in IG, however, were sustained for up to 6 months (*p* < 0.001 for each comparison of baseline to all other measurements). The results were similar for both the physical and psychological subscales of the MSIS-29, respectively.

### 3.3. Gait and Balance Measures

All parameters (10mWT, 2minWT, TS) improved significantly after inpatient rehabilitation in both treatment groups, apart from 10mWT in IG at discharge, where only a trend could be found (*p* = 0.098). Comparisons of month 3 vs. baseline were significant for both groups and all mobility outcomes, showing that rehabilitation effects could be maintained (Figure 4). 

From discharge to month 3, no significant changes for 10mWT and TS were seen in either group. The results for 2minWT, however, improved significantly during the 3 month internet program (Figure 4, bottom left), without significant changes in CG (Figure 4, top left). Data tables including all underlying numbers are available from the authors upon request.

### 3.4. Training Statistics and Adherence

Thirty-four participants analyzed in the IG recorded a total of 772 training sessions in the training software. The average number of training sessions per participant was 24.1 ± 17.7. Training frequency per week decreased continuously during the course of the intervention, ranging from 2.1 ± 1.9 per week in the first month to 1.4 ± 1.5 per week in the third month (overall average per participant was 1.7 ± 1.7). Mean duration (±standard deviation) of strength training sessions was 26.9 ± 10.1 min (range: 2–61), mean duration of endurance training sessions was 27.9 ± 20.3 min (range: 3–120). 

Therapists prescribed 140 different exercises during the course of the intervention. For strength training, the whole sample documented 3033 accomplished exercises (ae) in total. The most often performed exercises were leg muscle exercises in many variations (exercise categories: squats or lunges; ae = 949), lower back exercises (bridging/back extension, ae = 612), and upper back/shoulder exercises (elastic band pulling, ae = 608), followed by abdominal exercises (crunches/torso rotation, ae = 410), chest/arm exercises (pushups, elbow extension/flexion, ae = 199), core stability exercises (front/side planks, ae = 127), and ankle exercises (plantar flexion/extension, ae = 128). In addition, a total of 477 balance exercises (one-leg stance, standing on unstable surfaces) and 110 flexibility exercises (hip ae = 26, thigh ae = 51, torso rotation ae = 33) were documented. The most often performed endurance exercises were cycling (ae = 145), walking (ae = 113), cross-training (ae = 95), and running (ae = 26), totaling 379 accomplished endurance exercises. The average ratio of perceived exertion was 13.3 ± 1.8 for endurance exercises and 12.9 ± 2.0 for strength exercises. 

## 4. Discussion

The main finding of our study is that the positive effects of multimodal inpatient rehabilitation on MS-associated fatigue could be maintained with an individually administered, internet-based physical activity and exercise promotion program for 3 months—the predefined primary outcome measure. Moreover, these effects were sustained at the end of follow-up after 6 months, at a time point where no further therapist support had been given for 3 months. In parallel with the reduction of fatigue, HRQoL increased, as well as the walking distance in the 2 min walking test.

The observed effects on self-reported fatigue are remarkable given the fact that management of fatigue is difficult and treatment options are scarce [2]. Among these, exercise therapy has repeatedly been shown to ameliorate the subjective feeling of fatigue, as summarized in a Cochrane Review based on 26 randomized controlled trials that used a nonexercise CG [26], and a recent overview of Cochrane Reviews stating at least moderate-quality evidence for exercise and physical activity to reduce fatigue [27]. Our findings are in line with these data: both groups perceived less fatigue after inpatient rehabilitation, in which exercise and physical activity promotion were among the mainstays of therapies. 

Compared with baseline values, the median reduction at discharge of 15.5 and 18 points on the 0-68 point WEIMuS scale reflects an improvement of fatigue of about 34% and 46%, respectively. There is no information on minimal clinically relevant changes for the WEIMuS yet. For the Modified Fatigue Impact Scale, which is similar in nature, Learmonth et al. calculated that a change of 20.2 points (corresponding to a 24% change) is clinically relevant [28]. Thus, the magnitude of effects on fatigue in our study can be considered clinically meaningful. This holds true for both groups and provides evidence that inpatient rehabilitation may positively impact MS-associated fatigue. With no structured aftercare program, though, the positive effects of rehabilitation were nearly completely abolished three months afterwards. This means that a short-term intervention for fatigue lasting 3–4 weeks may not be sufficient to maintain the positive effects obtained during inpatient rehabilitation for longer periods of time, and this underscores the necessity for further studies with long-term follow-up periods, as has previously been advocated [29].

PwMS receiving the structured 12 week physical activity and exercise promotion program managed to maintain the positive effects of rehabilitation for 3 months. The implemented training frequency of the intervention (about two training units per week for 3 months) and the exercise session duration (about 25–30 min) and intensity (Borg ratio of perceived exertion of about 13) can be considered adequate to elicit those effects. 

Effects could even be preserved until 6 months after discharge, which means that fatigue levels were still reduced 3 months after cessation of the program. A plausible explanation for those long-term effects is that patients might have kept up their physical activity and exercise levels after the intervention, leading to stable fatigue reductions. This assumption is supported by the behavioral nature of the intervention that was aiming at increasing the physical-activity-related competences and autonomy that PwMS need to govern their own activity and training regime. Furthermore, internet-based aftercare programs can extend therapeutic support after rehabilitation into the living environment and everyday life of PwMS—right to where it matters the most for free-living activity promotion.

The time course of the MSIS-29 scores closely resembled that of the WEIMuS scores in both CG and IG. The MSIS-29 is an appropriate, reliable, and valid HRQoL instrument because it has been developed by rigorous psychometric methods [22], has better measurement properties and responsiveness than other HRQoL instruments [30], has been validated in different languages (including German) [31], and is thus a widely accepted method to measure the psychological and physiological impact of MS from the patient’s perspective [23]. On the one hand, our findings that exercise had an immediate effect on HRQoL (demonstrated during rehabilitation in both groups as well as during the home-based training in the IG) is in line with previous studies claiming that exercise can positively impact HRQoL [32]. On the other hand, the essentially similar time course of HRQoL corroborates the results of the WEIMuS scores and indirectly indicates the huge impact of fatigue on quality of life in PwMS [1,14,29]. 

At first glance, the course of mobility parameters seems to behave differently. As expected, and in accordance with the results of the patient-reported outcomes mentioned above, inpatient rehabilitation improved endurance (2 minWT), walking speed (10 mWT), and balance (TS). At follow-up after 3 months, the walking distance in 2minWT remained stable in the CG but increased further with the internet-based intervention, demonstrating that multimodal rehabilitation and exercise positively affects mobility, as has repeatedly been reported [27,33,34]. The fact that 10mWT and TS did not show further improvements after 3 months in the IG is not against that assumption because the performed training was mainly directed towards enhancing aerobic or muscular endurance, which is measured only with the 2minWT, and both tests showed ceiling effects that may have prevented further improvements.

Exercise is one of the most promising candidates not only to improve the various symptoms of MS, including fatigue and mobility (“tertiary prevention”), but also to influence the course of MS (“secondary prevention”) and possibly even to reduce the risk of developing MS or at least to postpone its manifestation (“primary prevention”) [35]. It has now become clear that, contrary to earlier beliefs, exercise and physical activity do not cause harm to PwMS and have a well-established safety profile without any relevant side effects, neither in terms of relapses nor in terms of persistent deterioration of MS symptoms [8,36]. Nevertheless, due to impairments in motor control and balance, severely affected patients might show increased risks of adverse events such as falling or stepping when exercising. The corresponding high demands on the individualization of exercise prescription could be met by the high number of available (220) and actually prescribed (140) different exercises in our study. In conjunction with the individual prescription, the rehabilitation experience of our highly trained therapists may have been the reason for the absence of exercise-related adverse events such as falls or injuries. Worthy of note here may be that in our study, possibly unlike in most other exercise studies in PwMS, the treating therapists were not physiotherapists but exercise therapists. Those therapists were especially trained in behavioral exercise therapy (as described elsewhere [16]), which was a cornerstone of our intervention. 

Despite the well-known beneficial effects and safety of exercise, the main challenge is (as in many other conditions and in healthy people as well) to motivate PwMS to engage in regular exercise. In this regard, internet-based approaches seem promising to promote physical activity in PwMS [11,37,38]. With the internet-delivered program used herein, we previously found that this type of exercise improved muscle strength, lung function, and physical activity but not fatigue [11]. The baseline fatigue score in this previous population (measured with the WEIMuS) was 21.3 ± 12.9, and thus markedly below the fatigue threshold of 32. In a second study using the internet-delivered program [21], we found that fatigue was only reduced in subgroups with either high fatigue or low aerobic capacity. This indicates that low fatigue scores do not leave much room for improvement, corresponding to a ceiling effect. In contrast, PwMS in the present study were primarily selected due to the presence of fatigue and were only included if the score on the WEIMuS scale indicated clinically relevant fatigue. This underlines the necessity of properly designed studies, with well-selected patient populations and a clear definition of the outcome of interest. 

There are some limitations to our study that merit discussion. At baseline, the IG showed significantly higher disease duration compared with the CG. Nevertheless, since there were no differences for disability (EDSS) and the study outcome parameters (MSIS-29, WEIMuS, 2minWT, 10mWT), the sample can be considered homogenous at baseline. An intention-to-treat analysis was not feasible since not all follow-up measurements could be carried out. The main reason for this was that a majority of participants had to travel long distances (some of which were several hundred kilometers) to the rehabilitation facility for the follow-up assessments. In line with our previous experience from internet intervention studies, some participants considered this too burdensome and therefore could not be included in the follow-up measurements. In total, 8 PwMS in the IG and 12 PwMS in the CG were lost to follow-up, resulting in a dropout rate of 19% and 29%, respectively. Interestingly, the dropout rate was higher in the CG than in the IG, and only one PwMS discontinued training due to technical problems. The dropout rate was similar to, or lower than, that for existing home training interventions for PwMS, which amounted to between 19% and 51% [39,40,41,42]. Besides dropout, adherence is a crucial aspect of exercise studies, even more so in internet-based interventions [10]. Like in previous studies [11,21], adherence (number of weekly or monthly exercise sessions documented) decreased during the course of the intervention period. This may have influenced the results. Therefore, the development of effective strategies to counteract diminishing adherence and training frequency is an important task for future research.

Our intervention followed a holistic approach, integrating individual exercise prescription and educational, behavior-oriented coaching in online (group) sessions. The contribution of each intervention element to the overall intervention effects measured at T2 and T3 cannot be determined. We did not influence physiotherapy utilization after inpatient rehabilitation and did not assess it at T2 or T3. Therefore, we do not know if there is a group difference for physiotherapy utilization that may have influenced the results. 

The aforementioned long travel distances to the study center were the reason why we conducted T3 assessments as a postal survey only, which may have influenced the results of this measurement time point. In addition, rater-based assessments were only available for up to 3 months and, therefore, the evolution of gait and balance measures beyond that time point has not been elucidated. Detailed information on physical activity during follow-up is missing and, therefore, we do not know whether the internet-delivered training had a long-lasting motivating effect and whether PwMS continued to exercise on their own during the last 3 months of our study. 

We could not perform genuine adverse event monitoring for all participants, as is common in pharmacological studies. Participants in the IG had continuous contact (messaging function, email) with their treating therapist and informally reported adverse events such as illness or relapses. For the CG, though, we do not have any information on adverse events. 

However, we believe that these limitations do not severely impact the main findings and that the limitations were outweighed by the strengths of our study: the randomized, controlled design; the well-selected patient population; the appropriate use of validated and established assessment instruments; the inclusion of sufficient numbers of PwMS in both groups; and the robust and consistent findings across the various outcome measures.

## 5. Conclusions

Taken together, the results of our study provide Class II evidence (according to [43]) that the effects of rehabilitation on fatigue, but also on quality of life and mobility parameters, can be maintained with an internet-delivered home exercise program for up to 3–6 months. Further studies with longer follow-up periods are needed to investigate whether such programs may have more sustainable effects, which patients benefit most, and what type and intensity of exercise is the most promising therapy in order to ameliorate the devastating consequences of fatigue in PwMS.

## Figures and Tables

**Figure 1 ijerph-17-04544-f001:**
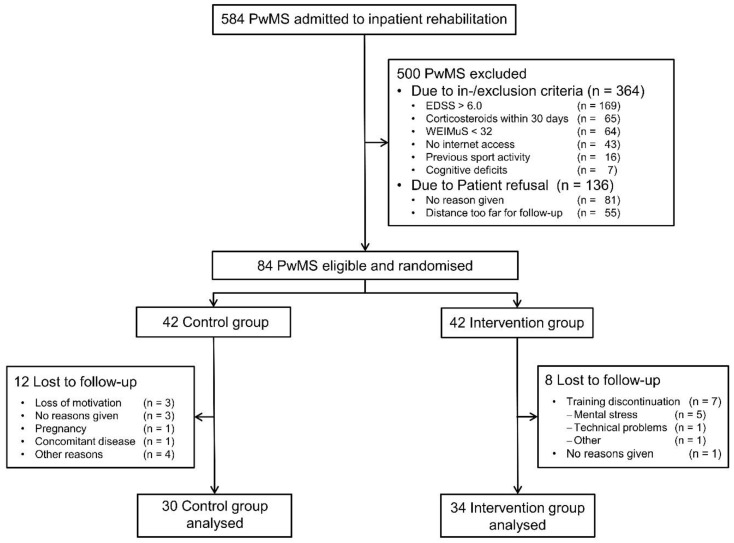
Flowchart of patient disposition. PwMS: persons with multiple sclerosis.

**Figure 2 ijerph-17-04544-f002:**
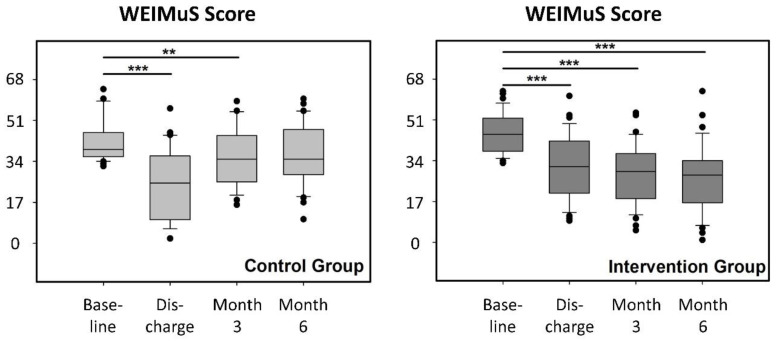
WEIMuS scores at baseline, discharge from rehabilitation, and after 3 and 6 months of follow-up after discharge in the control group (light grey) and intervention group (dark grey). Boxes: interquartile ranges; horizontal bar within the boxes: median values; whiskers: 10–90% ranges. Dots: individual values outside the 10–90% ranges. *** *p* < 0.001, ** *p* < 0.005 (Repeated Measures ANOVA on Ranks with Tukey’s test for pairwise comparisons).

**Figure 3 ijerph-17-04544-f003:**
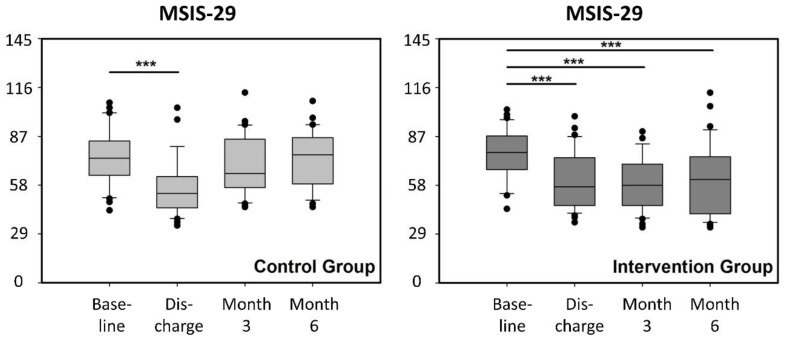
MSIS-29 scores at baseline, discharge from rehabilitation, and after 3 and 6 months of follow-up after discharge in the control group (light grey) and intervention group (dark grey). Boxes: interquartile ranges; horizontal bar within the boxes: median values; whiskers: 10–90% ranges. Dots: individual values outside the 10–90% ranges. *** *p* < 0.001 (Repeated Measures ANOVA on Ranks with Tukey’s test for pairwise comparisons).

**Figure 4 ijerph-17-04544-f004:**
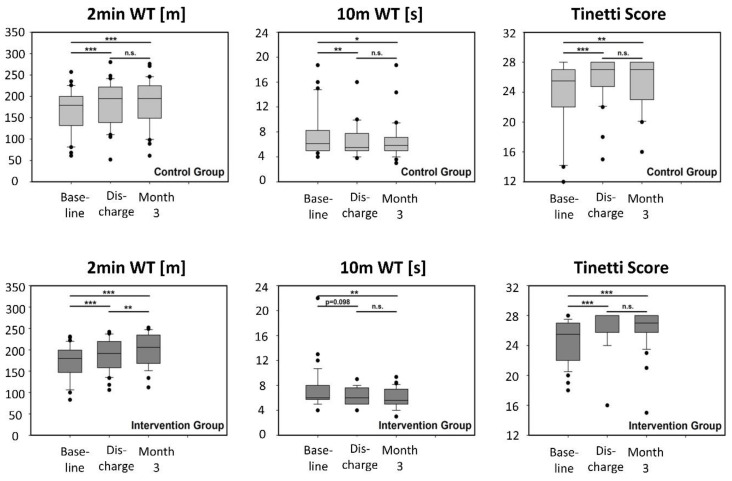
Gait and balance measures at baseline, discharge from rehabilitation, and after 3 months of follow-up after discharge in the control group (top, light grey) and intervention group (bottom, dark grey). 2min WT: 2 min walking test (meters); 10m WT: 10 m walking test (seconds); Tinetti score (ranging from 0 to 28). Boxes: interquartile ranges; horizontal bar within the boxes: median values; whiskers: 10–90% ranges. Dots: individual values outside the 10–90% ranges.*** *p* < 0.001, ** *p* < 0.01, * *p* < 0.02, n.s.: not significant (Repeated Measures ANOVA on Ranks with Tukey’s test for pairwise comparisons).

**Table 1 ijerph-17-04544-t001:** Measurement instruments, time points, and location/type. WEIMuS: Würzburg Fatigue Inventory for Multiple Sclerosis, MSIS-29: Multiple Sclerosis Impact Scale, 2minWT: 2 min walking test, 10mWT: 10 m walk test.

Measurements	Beginning of Inpatient Rehabilitation T0	End of Inpatient Rehabilitation T1	3 Months after Discharge T2	6 Months after Discharge T3
**Subjective fatigue** (WEIMuS)	Study center	Study center	Study center	Postal survey
**Quality of life** (MSIS-29)	Study center	Study center	Study center	Postal survey
**Gait** (2minWT, 10mWT)	Study center	Study center	Study center	/
**Balance** (Tinetti Score)	Study center	Study center	Study center	/

**Table 2 ijerph-17-04544-t002:** Demographics and baseline characteristics of the intervention and control groups. EDSS: Expanded Disability Status Scale, WEIMuS: Würzburg Fatigue Inventory for Multiple Sclerosis, MSIS-29: Multiple Sclerosis Impact Scale, 2minWT: 2 min walking test, 10mWT: 10 m walk test, RR MS: relapsing remitting MS, SD: standard deviation, IQR: interquartile range. *p*-Values are given for differences between groups (Mann–Whitney rank sum test, or Chi-Square test if indicated with an asterisk *).

	Intervention Group (n = 34)	Control Group (n = 30)	*p*-Value
**Age** [years] mean ± SD	47.6 ± 9.2	46.4 ± 12.2	0.328
**Women** [n] (percent)	22 (65%)	17 (57%)	0.279 *
**Disease duration** [years] mean ± SD	13.4 ± 7.9	9.0 ± 7.5	0.015
**EDSS** median (IQR)	4.3 (3.5–5.0)	4.0 (3.0–6.0)	0.828
**RR MS** [n] (percent)	19 (56%)	20 (67%)	0.531 *
**WEIMuS** median (IQR)	45 (38–52)	39 (36–46)	0.13
**MSIS-29** median (IQR)	77.5 (67.3–87.3)	74.0 (63.8–84.3)	0.492
**2minWT** median (IQR)	179.5 (147.0–199.3)	179.0 (131.8–200.0)	0.619
**10mWT** median (IQR)	6.65 (5.85–8.03)	6.90 (5.60–8.98)	0.777

**Table 3 ijerph-17-04544-t003:** Differences in WEIMuS scores from baseline to discharge, month 3, and month 6 in the intervention and control groups. Data are given as median, the numbers in brackets indicate the 25–75% interquartile range (IQR). *** *p* < 0.001, IG vs. CG, Mann–Whitney rank sum test.

	Intervention Group (n = 34)	Control Group (n = 30)
**Baseline–Discharge**	15.5 (5–22)	18 (11–28)
**Baseline–Month 3**	16.5 *** (10–29)	7 (2–11)
**Baseline–Month 6**	22.5 *** (8–30)	5.5 (1–11)

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
