# Peer review of "Efficacy of an Internet-Based Program to Promote Physical Activity and Exercise after Inpatient Rehabilitation in Persons with Multiple Sclerosis: A Randomized, Single-Blind, Controlled Study"

_ijerph, 2020, doi:10.3390/ijerph17124544_

Round 1

Reviewer 1 Report

Line 101 - subjects had free choice of activity for 10-60 minutes. Were there any choice limitations? What were the activities chosen? If you have the information, please include numbers of subjects for each activity. Example: stationary bike (n), street bike (n), treadmill walking (n), indoor walking (n), outdoor walking (n), swimming (n), etc.

In section 3.4 - You report 772 training sessions. What was the average duration and range of the training sessions? Ideally you would report this separately for resistance exercise and endurance exercise and then report the combined duration.

Line 311 needs a grammatical correction.

Line 320 - You want more research for more information on type of exercise that is most promising. Adding the details mentioned above will make this paper more useful to other researchers.

Well designed study and report.

Author Response

Dear Reviewer,

Thank you for your insightful and helpful comments! You are absolutely right in saying that more information on prescribed and accomplished exercises would be helpful for readers and future research. Therefore, we added extensive statistical details about documented exercises, duration of exercise sessions and intensity. We were able to successfully address your comments (see below) as follows (our comments italic and green):

  1. Line 101 - subjects had free choice of activity for 10-60 minutes. Were there any choice limitations? What were the activities chosen? If you have the information, please include numbers of subjects for each activity. Example: stationary bike (n), street bike (n), treadmill walking (n), indoor walking (n), outdoor walking (n), swimming (n), etc.

    There were no limitations for choice of activity. To illustrate the chosen activities, we included in depth information about documented exercises in chapter 3.4., which was renamed to “training statistics and adherence”

  2. In section 3.4 - You report 772 training sessions. What was the average duration and range of the training sessions? Ideally you would report this separately for resistance exercise and endurance exercise and then report the combined duration.

    We also added information about the duration (mean, standard deviation, range) of strength training and endurance training sessions in chapter 3.4

  3. Line 311 needs a grammatical correction.

    We corrected that sentence

  4. Line 320 - You want more research for more information on type of exercise that is most promising. Adding the details mentioned above will make this paper more useful to other researchers.

    You are absolutely right, and we dare to go even further in that direction. Alongside with type of activity, the intensity may also be of interest. Therefore, we added information about the mean subjective ratio of exertion of endurance and strength exercises in chapter 3.4 and added an evaluation of session duration and exercise intensity into the discussion (lines 327-328).

  5. Well designed study and report.

    Thank you for your appreciation

Reviewer 2 Report

I have read with interest the manuscript entitled “Efficacy of an internet-based program to promote physical activity and exercise after inpatient rehabilitation in persons with multiple sclerosis: a randomized, single-blind, controlled study. in general terms I think it is a good job, however, it needs to improve a little to be published in IJERPH

Major comments

Please, in methods describe the procedure to measure the adherence

Please, describe with more details the program of exercise. For example, what muscles are involved the strength training?. In the endurance training, what are the exercises? Walk, bicycle,?. Does the programme include warming or flexibility exercises?.

The authors did not determine the sample size. I suppose it is because this work is a continuation of another. They could then at least calculate the power of the results.

In the table of results, provide the p-value for the comparison between baseline characteristics.  Add the Tinetti, 10m WT, 2min WT, MSIS 29 for this analysis. It is essential to describe if the populations were homogeneous before the intervention.

The authors measured the training sessions and the compliance, and describe the diminution during the intervention. However, these results do not have been discussed. Please, add a paragraph about how adherence can affect the results. This is an essential point in a telemedicine intervention.

Author Response

Dear Reviewer,

thank you for your helpful comments regarding intervention details, methods, results and discussion! We were able to successfully address your comments (see below) as follows (in green and italic):

  1. Please, in methods describe the procedure to measure the adherence

    We added an explanation in the methods section on how we analysed adherence (lines 131-133)

  2. Please, describe with more details the program of exercise. For example, what muscles are involved the strength training?. In the endurance training, what are the exercises? Walk, bicycle,? Does the programme include warming or flexibility exercises?.

    We added extensive information about the choice of exercises by therapists, the provided exercise catalogue and special considerations for more severely disabled pwMS in chapter 2 (lines 121ff). In the results section in chapter 3.4 (lines 281ff), we added information on number and type of exercises performed, perceived ratio of exertion and session duration. Flexibility exercises could be prescribed on demand, as stated now in the methods (lines 124ff) and in chapter 3.4. We added information that no standardized warmup was performed (line 122).

  3. The authors did not determine the sample size. I suppose it is because this work is a continuation of another. They could then at least calculate the power of the results.

    This is a very interesting point. Sample size calculation indeed had been difficult since, to our knowledge, ours was the first study evaluating internet-based exercise promotion after rehabilitation that targets subjective fatigue. Thus, no information about expected effect sizes and statistical power was available, and we indeed assumed from previous studies that a sample size of about 80 is sufficient. This assumption was supported by an ex post power calculation for the primary outcome which we added as requested (lines 189-191).

  4. In the table of results, provide the p-value for the comparison between baseline characteristics.  Add the Tinetti, 10m WT, 2min WT, MSIS 29 for this analysis. It is essential to describe if the populations were homogeneous before the intervention.

    As requested, we added the additional outcomes to table 1 and calculated p-values for differences between groups. The only significant baseline difference was found for disease duration, which was higher for the intervention group, but did not translate to disability (EDSS) or outcome parameters (MSIS-29, WEIMuS, mobility). Therefore, the sample can be considered homogenous at baseline. We added a statement to the results (lines 198-200) and discussion (lines 388-391)

  5. The authors measured the training sessions and the compliance, and describe the diminution during the intervention. However, these results do not have been discussed. Please, add a paragraph about how adherence can affect the results. This is an essential point in a telemedicine intervention.

    We added a paragraph in the discussion/limitation (lines 400-405), addressing adherence issues

Reviewer 3 Report

This study addresses an interesting area such as therapeutic exercise in patients with Multiple Sclerosis. However, the measuring instruments are based on scales that do not really measure the maintenance of functionality. To truly assess the influence of the intervention on achieving a less cost-energy gait, kinematic or electromyographic studies should be implemented that could correlate exercise with decreased fatigue (for example, maximum voluntary contraction test), maintaining the achievements achieved in rehabilitation.

Page 1-21. Patients with EDSS 5-6 were not assisted at the risk of falls? his type of patient can be included in face-to-face therapy without risk, in telerehabilitation it would be difficult if they did not have assistance.

Page 1-21. It is necessary to indicate in the Abstract the number of subjects assigned to each group. What type of intervention did the control group perform? What does "usual care" mean?

Page 1-24. Replace "mobility" with "gait".

Page 1-25. "At the end of rehabilitation" means at the end of inpatient rehabilitation? Specify.

Page 1-25 None of the variables described in the Abstract were measured in T3, only HRQoL, as described in the body of the manuscript.

Page 1-27 There is no mention in the manuscript that MSIS-29 was measured at T3.

Pag 1-29 “Gait parameters” under research refers to parameters such as cadence, step length, etc. that in this study were not measured. Furthermore, it is not specified at what time it improved (T1, T2, ...), it being unclear if the improvement was due to impatient rehabilitation.

Page 1-32. Check that Keywords are MeSH terms.

Page 2-59 "Exercise therapists" refers to "Physical Therapists? The health professional in charge of prescribing therapeutic exercise to neurological patients is the physical therapist.

Page 2-92 Was the individual and group intervention equally effective?

Page 2-70 The evaluation at 6 months was done through a "postal survey"? Could this alter the results?

Page 3-99. Did the sports professionals maintain contact with the physical therapists who cared for the patients during the rehabilitation process? Perhaps some of the proposed exercises could go against the prescriptions of the physical therapists, regarding specific deficits such as weakness, dropped foot, etc. which could increase the risk of falls. The physical therapy intervention from T0 to T1 was not described and is a fundamental aspect in the patient’s recovery.

Page 3-107. If a pre-post impatient rehabilitation was also carried out, should it not be reflected as another of the objectives of the study?

Page 3-111. Only one HRQoL measuring instrument was measured at 6 months?

Page 4-154. Figure 1 should appear in the Methodology section.

Page 5-162. In Methodology it is not described that this scale was measured in T3.

Page 5-169. The great variability in T3 in the intervention group, both in Fig 1 and in Fig. 2, is striking. What could it have been? Could not the difference in modality in passing the scales in T0, T1 and T2 in person and in T3 by means of a “postal survey” bias the results?

Page 5-182. Should not a T0-T1, T1-T2 and T1-T3 analysis have been performed, since the intervention during T0-T1 was different from that of T1-T3?

Page 6-196. Modify “mobility parameters, gait is really being evaluated.

Page 7-222. It was not specified in the Methodology section that from T2 to T3 the patients were not supervised.

Page 8-239 The structure aftercare program should not be carried out by health professionals such as physical therapists who implement a rehabilitation o telerehabilitation program? It is important to take into account important factors on gait performance, not only on gait speed when individualizing Telerehabilitation programs.

Page 8-267. Add a table to observe to appreciate these numerical changes, as in the other variables. These changes are not well seen in Fig 4.

Page 8-283. The bibliography refers to “the relative risk of relapse associated with physical exercise”, however, it would be necessary to report the risk of falling in performing the exercises when the person trains at home without the assistance of a physical therapist, especially in those with greater grade of disability.

Page 9-307. Precisely the evaluation of the gait could be a bit more objective. Completing a questionnaire could have been highly conditioned because the patients knew which group they were in, since double-blind was not possible. More objective evaluation systems would be necessary to overcome this difficulty in T3.

Page 9-308. The maintenance of improvement at 6 months is striking, despite not making a face-to-face or supervised intervention in this period, could the subjectivity of the aforementioned survey influence?

Author Response

Dear Reviewer,

Thank you for your time and effort in giving us helpful comments! There were major considerations and numerous “small” observations we had overseen that really helped us to improve and clarify the manuscript. We successfully implemented your comments as indicated in the point-by-point comments below.

Some general statements first:
You are absolutely right in saying that energy cost of walking, or kinematic or electromyographic outcomes would have been an additional asset to our study. Nevertheless, given the character of a pilot study and limited resources, we were not able to conduct spiroergometric analyses or any electrophysiological measures. In addition, our focus was not on motor aspects of fatigue but on subjective aspects, which seem not to be related to motor fatigue/fatigability. The WEIMuS questionnaire is a validated and frequently used patient-reported outcome to assess subjective fatigue in persons with MS.  

Moreover, thank you for your concerns about the potential danger of falls in severely affected patients. We have extensive experience from previous studies in internet-based exercise prescription in persons with MS (about 500 treated patients). Among those studies, there are two which recruited only severely affected patients with EDSS up to 7.0 including 75 patients in total (Frevel & Mäurer Eur J Phys Rehabil Med 2015; 51:23-30; https://clinicaltrials.gov/ct2/show/NCT03548974). According to our experience, telerehabilitation is feasible in this population, provided that therapists are highly trained, exercise prescription is highly individualized and safety precautions are met. So far, we did not have exercise-related adverse events in our study populations. We added additional information related to this issue in the manuscript as indicated below.

Another major point raised in the review is the fact that, after 6 months, we only conducted a postal survey. The reasons were pragmatic: as stated in lines 392-396, participants had long travel distances, and imposing long travels again at T3 would have led to a considerable dropout rate. This may indeed have biased the results. To our knowledge, there is no study which analysed any potential bias in persons with MS regarding the environment where questionnaires are filled out. Thus, there are no existing signs or hints for the existence of such a bias, yet we cannot rule out that there is a bias. 

please find your comments below, and our responses (green and italic). 

  1. Page 1-21. Patients with EDSS 5-6 were not assisted at the risk of falls? his type of patient can be included in face-to-face therapy without risk, in telerehabilitation it would be difficult if they did not have assistance.

We added several paragraphs explaining exercise prescription, safety precautions and therapist qualification in the manuscript (lines 124ff, 366ff)

  1. Page 1-21. It is necessary to indicate in the Abstract the number of subjects assigned to each group. What type of intervention did the control group perform? What does "usual care" mean?

In Germany, care after rehabilitation is not standardized. Usually, there is either no intervention or physiothearapy; both depends on the discretion o fthe family practicioner and/or the neurologists. Obviously, this is country-specific and needs to be clarified. We replaced usual care with “physiotherapy/no intervention” in the abstract (line23) and added an additional statement in lines 134-138. We added the number of participants in CG and IG in line 26.   

  1. Page 1-24. Replace "mobility" with "gait".

We did as was requested. 

  1. Page 1-25. "At the end of rehabilitation" means at the end of inpatient rehabilitation? Specify.

Yes, we are referring to inpatient rehabilitation and added “inpatient” in line 25.

  1. Page 1-25 None of the variables described in the Abstract were measured in T3, only HRQoL, as described in the body of the manuscript.

You are right, we did not report all outcomes at all measurement time points in the abstract. This was due to word count restrictions (200 words limit is quite demanding). We reported fatigue and MSIS-29 for T3, though (lines 27 and 28).  

  1. Page 1-27 There is no mention in the manuscript that MSIS-29 was measured at T3.

Yes, there is. To make measurements and time points clearer and more intelligible, we added table 1 instead.  

  1. Pag 1-29 “Gait parameters” under research refers to parameters such as cadence, step length, etc. that in this study were not measured. Furthermore, it is not specified at what time it improved (T1, T2, ...), it being unclear if the improvement was due to impatient rehabilitation.

Yes, this may be misleading. We changed “gait parameters” to just “gait”. We specified the time point for the observed gait improvements in line 29.

  1. Page 1-32. Check that Keywords are MeSH terms.

We replaced “mobility” by “walking”, “exercise promotion” by “exercise”, “internet intervention” by “internet-based intervention”, “e-health” by “online systems” and “behavior change” by “health behavior” to assure that all keywords are also MeSH terms.    

  1. Page 2-59 "Exercise therapists" refers to "Physical Therapists? The health professional in charge of prescribing therapeutic exercise to neurological patients is the physical therapist.

You are making an interesting point here. In Germany, there is a clear distinction between physiotherapists (non-academic) and exercise therapists (in general academic, e.g. sports science, sports therapy, rehabilitation science). Both professions have different functions in the rehabilitation process. Exercise therapists are usually responsible for group therapy, training/exercise therapy, patient education and behavioral exercise therapy. Since exercise therapists are better educated in training science, exercise counselling and control, and behavior change, it is quite natural that we chose exercise scientists to care for our patients online. The term “exercise therapist” may lead to internationally diverging interpretations, nevertheless there is – to our knowledge – no better translation.

We added a statement concerning the qualification and experience of our therapists in line 129.  

  1. Page 2-92 Was the individual and group intervention equally effective?

If we interpret this question right, you are asking if we can determine how much the individual exercise prescription and the group intervention contributed to the overall intervention effects? Well, we can´t. We added a corresponding statement into the discussion/limitation section. (lines 406-408).   

  1. Page 2-70 The evaluation at 6 months was done through a "postal survey"? Could this alter the results?

We cannot rule this out, and added a corresponding statement in lines 409-411.

  1. Page 3-99. Did the sports professionals maintain contact with the physical therapists who cared for the patients during the rehabilitation process? Perhaps some of the proposed exercises could go against the prescriptions of the physical therapists, regarding specific deficits such as weakness, dropped foot, etc. which could increase the risk of falls. The physical therapy intervention from T0 to T1 was not described and is a fundamental aspect in the patient’s recovery.

The precise contents of the multimodal rehabilitation are individualised and as well country-specific, and international comparability indeed is a problem. Nevertheless, it is hardly possible to describe inpatient rehabilitation in more detail than in lines 95-96. You are right, in a best case scenario the inpatient physiotherapists establish close connection to aftercare interventions. Unfortunately, the German health system gives very small room for those scenarios. Thus, inpatient rehabilitation and internet-based aftercare took place separate from each other, reflecting the reality of care. Online therapists conducted, among others, an extensive interview assessing medical history and goals to be able to individually tailor the intervention and reduce adverse events. This is described in detail in [20], for immediate information for the reader we added information about the qualification of our therapists and prescription details (lines 121-130), and about the absence of exercise-related adverse events (lines 366-374).

  1. Page 3-107. If a pre-post impatient rehabilitation was also carried out, should it not be reflected as another of the objectives of the study?

When designing the study and after some considerations, we opted against this strategy. We did not focus on the effects of inpatient rehabilitation because those are quite well confirmed. There is no information, however, on the potential of internet-based rehabilitation aftercare on the maintenance or even extension of rehabilitation effects over time. In addition, establishing a randomized, controlled situation during rehabilitation would have been hardly feasible, and we wanted to establish a scientifically rigid and sound study design, for both primary and secondary outcomes. Nevertheless, there were significant differences in the pre-post comparison in both groups and thus, rehabilitation was effective.     

  1. Page 3-111. Only one HRQoL measuring instrument was measured at 6 months?

At 6 months, HrQoL and fatigue were assessed, as at the other time points. We admit that the narrative description of measurement time points, measurement instruments and mode (study center vs. postal survey) is confusing. We added a tabular view (new table 1) that should make things clearer. Thanks for this comment!

  1. Page 4-154. Figure 1 should appear in the Methodology section.

We know that this is not handled homogeneously. With regard to the design of the flowchart we referred to CONSORT standards (Consolidated Standards of Reporting Trials,  (http://www.consort-statement.org/) and followed also the CONSORT recommendation to insert the flowchart as first element of the results section. Since CONSORT flowcharts also contain dropouts, which rather belong to results than design of a study, we would argue to keep the flow chart within the results section.  

  1. Page 5-162. In Methodology it is not described that this scale was measured in T3.

This is solved by the new table 1 displaying all measurement instruments and time points.

  1. Page 5-169. The great variability in T3 in the intervention group, both in Fig 1 and in Fig. 2, is striking. What could it have been? Could not the difference in modality in passing the scales in T0, T1 and T2 in person and in T3 by means of a “postal survey” bias the results?

We hope to interpret this right, since Fig 1 is the flow chart, and Fig 2 is the WEIMuS chart. We assume that you are referring to Fig 2 (WEIMuS) and Fig 3 (MSIS-29)?
Honestly speaking, we cannot see any striking variability in Fig 2. The vertical height of the WEIMuS T3-Box of the intervention group is the same size or even smaller than at T1 and T2 of IG, and T1, T2, and T3 of the CG. We agree, though, that the position of the outliers at T3 of the IG has a larger vertical spread. This is caused by two outliers only, so conclusions or interpretations would lack a statistical basis. We see an increase in variability after inpatient rehab at T1, though, which probably is due to individual variation of treatment response.

In Fig 3, the vertical height of the T3-Box and the vertical span of the outliers indeed appears to be higher than the T2 or T1 boxes. Whether this can be interpreted statistically or causally remains speculative. Since we cannot rule out that the postal character of T3 may have contributed, we added a statement that the postal survey at T3 may have influenced the results (lines 409-411).

  1. Page 5-182. Should not a T0-T1, T1-T2 and T1-T3 analysis have been performed, since the intervention during T0-T1 was different from that of T1-T3?

This is an interesting aspect. As explained above, we opted against an evaluation of inpatient rehabilitation, and focussed on the potential of internet-based rehabilitation aftercare to sustain rehabilitation effects. Thus, we were interested in the total effect of the “treatment unit” [rehabilitation + aftercare] and the maintenance of its effects compared to baseline.

  1. Page 6-196. Modify “mobility parameters, gait is really being evaluated.

We exchanged “mobility parameters” for “gait and balance measures” in that line and in line 171

  1. Page 7-222. It was not specified in the Methodology section that from T2 to T3 the patients were not supervised.

We added a short information in line 97, and additions in line 134-138 to clarify this.

  1. Page 8-239 The structure aftercare program should not be carried out by health professionals such as physical therapists who implement a rehabilitation o telerehabilitation program? It is important to take into account important factors on gait performance, not only on gait speed when individualizing Telerehabilitation programs.

If we interpret your question right, you are saying that the physiotherapists who were in charge in inpatient rehabilitation should also conduct the telerehabilitation intervention afterwards, because they have information about the patient´s quality of gait? Generally, we absolutely agree on that.  Nevertheless, as we explained above, this is hardly feasible within the German healthcare system. In order to reflect reality of care, we used different health care professionals for the internet-based intervention after inpatient rehabilitation. Due to the academic background, exercise therapists are well qualified to design and implement a telerehabilitation program (at least in Germany). Of course, extensive experience in the rehabilitation setting and individual exercise prescription is a prerequisite for high quality of care and the avoidance of adverse events. We clarified this in the manuscript (lines 124-130, 366-374).

Information about qualitative aspects of gait would have been a valuable addition to our study. Since we did not focus on motor fatigue but on subjective aspects of fatigue, and due to the pilot character of our study, we did not implement those measurements. 

  1. Page 8-267. Add a table to observe to appreciate these numerical changes, as in the other variables. These changes are not well seen in Fig 4.

Due to the numerous aspects of data displayed in this graph, we deliberately opted against using a table. We are of the opinion that a table could not very well handle all data layers and aspects in the graph: time points (T0, T1, T2), group (IG, CG), significance of differences and p-value (T0 vs T1, T0 vs T2, T1 vs T2), measurement instruments (2minWT, 10mWT, Tinetti), results characteristics (median, 25% - 75% Interquartile ranges, outliers). In our opinion, a table would get too bulky (even more so for secondary outcomes). Therefore, we argue to keep the figure.    

  1. Page 8-283. The bibliography refers to “the relative risk of relapse associated with physical exercise”, however, it would be necessary to report the risk of falling in performing the exercises when the person trains at home without the assistance of a physical therapist, especially in those with greater grade of disability.

As stated above, we took precautions for the safety of our participants, and the absence of adverse events proves the feasibility and safety of our approach even in more disabled pwMS. Corresponding statements were added to the manuscript (see comments above). Thus, we added the requested information about the occurrence of falls during exercise. We do not have any information about whether our intervention influences the general risk of falling, unfortunately, since this was out of the scope of our study.  

  1. Page 9-307. Precisely the evaluation of the gait could be a bit more objective. Completing a questionnaire could have been highly conditioned because the patients knew which group they were in, since double-blind was not possible. More objective evaluation systems would be necessary to overcome this difficulty in T3.

The gait and balance measurements collected at T0, T1 and T2 were all alike and objective. You are right, we also would have preferred to extend those measurements to T3, but this was not feasible. We added an explanation in lines 392-396 and 409-411. Here, we also admitted that the absence of gait and balance measures at T3 is a limitation.

  1. Page 9-308. The maintenance of improvement at 6 months is striking, despite not making a face-to-face or supervised intervention in this period, could the subjectivity of the aforementioned survey influence?

As stated above, there is- to our knowledge - no evidence for a possible bias regarding the situation a questionnaire is filled in (on site vs. at home). Since we cannot rule this out, we added the corresponding statement (line 409-411)

Round 2

Reviewer 3 Report

Thank you very much for the aclarations. Below, I detail some points that should still be clarified:
1, 2 and 13. Could the control group then perform Physiotherapy or no intervention from T1 to T2? This could greatly influence the results, and it is necessary to report it as a limitation. The interventions would not be homogeneous within the group, and thus the groups would not be comparable.
9. Thanks for the clarification. Given that it can cause confusion for most readers internationally and, even if it is not a literal translation, would it be possible to report that the intervention was not carried out by physical therapists and this could have influenced the results?
17. Sorry for the error, I was referring to Table 1 and 2, now Table 3 (22.5 - 5.5).
18. It would be interesting to include it in the Discussion.
9 and 21. Academic baggage undoubtedly influences the good design of a study and its interpretation. However, the professionals who carry out the interventions must first and foremost have training and years of clinical experience, as well as specific instructions on the protocols to be implemented. In fact, in most studies it is said that the intervention was carried out by physiotherapists with at least x years of experience to ensure that the interventions are optimal. It would be interesting to include non-measurement of gait kinematics as a limitation.
22. Add it to the limitations of the study.

Author Response

Dear Reviewer,

thank you for your additional comments and remarks.

We adressed the points you raised as follows (our answers in italic and green color).

Point-by-point response:

  • 1, 2 and 13. Could the control group then perform Physiotherapy or no intervention from T1 to T2? This could greatly influence the results, and it is necessary to report it as a limitation. The interventions would not be homogeneous within the group, and thus the groups would not be comparable.

The physiotherapy utilization of any of our study participants between T1 and T3 was neither restricted nor facilitated. This reflects usual care in Germany, since after rehabilitation there is no standardized aftercare. For our study, this means that participants in the IG and CG had the same probability to utilize physiotherapy services or not. Thus, the groups are comparable in this regard. It is a limitation, though, that we did not assess physiotherapy utilization at T1 or T2, so we do not know if there might have been difference at random between groups.

We made this clearer in the manuscript. Due to the word count limit in the abstract, and trying not to confuse readers early, we just stated that the CG did not receive any intervention (line 23). In lines 137-138, we added “Health care utilization between T1 and T3, including physiotherapy, was neither facilitated nor restricted for IG and CG alike.” In lines 410-412, we added “We did not influence physiotherapy utilization after inpatient rehabilitation, and did not assess it at T2 or T3. Therefore, we do not know if there is a group difference for physiotherapy utilization that may have influenced the results.”

  • Thanks for the clarification. Given that it can cause confusion for most readers internationally and, even if it is not a literal translation, would it be possible to report that the intervention was not carried out by physical therapists and this could have influenced the results?

We agree that the fact that our intervention was delivered by exercise therapists and not physiotherapists is worth mentioning. Therefore, we added the following statement in lines 375-377: “Worthy of note here may be that in our study, possibly unlike in most other exercise studies in PwMS, treating therapists were not physiotherapists but exercise therapists. Those therapists were especially trained in behavioural exercise therapy (as described elsewhere [16]), which was a cornerstone of our intervention”.

  • Sorry for the error, I was referring to Table 1 and 2, now Table 3 (22.5 - 5.5).

Thanks for the clarification. We already solved this issue then, during the last review process, by adding a statement regarding the potential bias due to the postal survey of T3 (lines 415-417).

  • It would be interesting to include it in the Discussion.

It might possibly be interesting to discuss other study design options. Nevertheless, we prefer to discuss the chosen design alone, and its strengths and weaknesses. Discussion of alternate design options may be confusing to the reader and may dilute our line of argumentation. 

  • 9 and 21. Academic baggage undoubtedly influences the good design of a study and its interpretation. However, the professionals who carry out the interventions must first and foremost have training and years of clinical experience, as well as specific instructions on the protocols to be implemented. In fact, in most studies it is said that the intervention was carried out by physiotherapists with at least x years of experience to ensure that the interventions are optimal. It would be interesting to include non-measurement of gait kinematics as a limitation.

We agree that treating therapists need to have extensive clinical experience to be able to treat persons with MS and the myriad of symptom constellations adequately. Thus, we happily added statements about our therapists’ extensive clinical experience during the last review process. In the statement added to lines 375-377 (see above), we now also stated that our therapists were trained for our intervention concept and protocol.

According to a recent meta-analysis, motor fatigue (fatigability) and subjective fatigue are related, but two different constructs (Loy et al. 2017, Journal of psychosomatic research). As explained, we focussed exclusively on subjective fatigue. Measuring fatigability by kinematic gait assessments would undoubtedly have been interesting, but also a separate research question. Therefore, we disagree that the absence of kinematic gait assessments is a limitation to our study. It is not a limitation to our protocol, our outcomes the interpretation of our results. Thank you for your understanding.

  • Add it to the limitations of the study.

We consider this issue maybe not placed correctly within the limitations. Instead, we suggest an additional statement “Data tables including all underlying numbers are available from the authors upon request” (lines 276-277).  
